# Automatic Stones Classification through a CNN-Based Approach

**DOI:** 10.3390/s22166292

**Published:** 2022-08-21

**Authors:** Mauro Tropea, Giuseppe Fedele, Raffaella De Luca, Domenico Miriello, Floriano De Rango

**Affiliations:** 1Department of Informatics, Modeling, Electronics and Systems Engineering (DIMES), University of Calabria, Via P. Bucci, 87036 Rende, Italy; 2Department of Biology, Ecology and Earth Sciences (DiBEST), University of Calabria, Via P. Bucci, 87036 Rende, Italy

**Keywords:** *Deep Learning* (DL), *Convolutional Neural Network* (CNN), *Machine Learning* (ML), *Softmax*, *Support Vector Machine* (SVM), *k-Nearest Neighbors* (kNN), *Random Forest* (RF), *Gaussian Naive Bayes* (GNB), Two-Stage Hybrid Model

## Abstract

This paper presents an automatic recognition system for classifying stones belonging to different Calabrian quarries (Southern Italy). The tool for stone recognition has been developed in the SILPI project (acronym of “*Sistema per l’Identificazione di Lapidei Per Immagini*”), financed by POR Calabria FESR-FSE 2014-2020. Our study is based on the *Convolutional Neural Network* (CNNs) that is used in literature for many different tasks such as speech recognition, neural language processing, bioinformatics, image classification and much more. In particular, we propose a two-stage hybrid approach based on the use of a model of *Deep Learning* (DL), in our case the CNN, in the first stage and a model of *Machine Learning* (ML) in the second one. In this work, we discuss a possible solution to stones classification which uses a CNN for the feature extraction phase and the *Softmax* or *Multinomial Logistic Regression* (MLR), *Support Vector Machine* (SVM), *k-Nearest Neighbors* (kNN), *Random Forest* (RF) and *Gaussian Naive Bayes* (GNB) ML techniques in order to perform the classification phase basing our study on the approach called *Transfer Learning* (TL). We show the image acquisition process in order to collect adequate information for creating an opportune database of the stone typologies present in the Calabrian quarries, also performing the identification of quarries in the considered region. Finally, we show a comparison of different DL and ML combinations in our Two-Stage Hybrid Model solution.

## 1. Introduction

In the course of evolution, humans have developed complex skills to adapt to the surrounding environment and act on the basis of what has been observed. Depending on the situation, we are able to decide the most appropriate behavior to use according to a certain pattern, which can be, for example, recognizing a face, understanding another person’s words, reading handwriting or distinguishing fresh food from its smell. The development of technology and the exponential improvement of computational sciences have made it possible to create computer learning software. This software acts by recognizing a certain scheme, depending on the application. *Pattern Recognition* (PR) is a branch of *Artificial Intelligence* (AI) that focuses on the recognition of patterns, forms and classifications in data by a computer. It is closely related to *Machine Learning* (ML), data mining and the discovery of knowledge. It aims to classify objects into a number of categories or classes. The main phase of a PR process concerns the *Feature Extraction and Classification*. Its goal is to characterize the data to be recognized by metrics that will provide the same results for the data in the same category and different results for the data in different categories. This leads to finding distinctive features that are invariant to any data transformation (ideally). The degree of classification of the input into different categories varies according to the characteristics of the data. In this work, we used the *Convolutional Neural Network* (CNN) to perform the feature extraction phase which is one of the most important steps in the PR, and the TL [1] approach to avoid creating our network from scratch. In particular, we used a Two-Stage Hybrid Model solution that joins the use of a *Deep Learning* (DL) technique, a CNN model for the feature extraction phase, with a classical ML algorithm in order to perform image classification. We used four different CNNs, each one implementing five types of ML algorithms for classification: the *Softmax* or *Multinomial Logistic Regression* (MLR) [2], the *Support Vector Machine* (SVM) [3], the *k-Nearest Neighbors* (kNN) [4], the *Random Forest* (RF) [5] and the *Gaussian Naive Bayes* (GNB) [6]. We have obtained the confusion matrix of the performed object recognition for each type of used algorithm. Finally, we have presented a comparison between these algorithms in order to show the performances of each approach. In this scenario, the contribution of this paper is to give some indications into the development of a system for automatic stones classification from the Calabrian quarries. The tool for stone recognition has been developed in the SILPI project (acronym of “*Sistema per l’Identificazione di Lapidei Per Immagini*”), financed by POR Calabria FESR-FSE 2014-2020 [7]. The characterization and the determination of the provenance of stone materials, generally, represent a very long and complex process that requires not only the use of destructive and expensive diagnostic techniques, but also a specialized staff with scientific and technical know-how who are able to interpret and process the compositional data obtained from the analyses. Instead, the system developed in this project is intended to be a tool that can be easily used by non-geologists (such as restorers, archaeologists, architects, engineers, diagnostics and art historians) by helping them to solve problems about the provenance and the classification of stone materials. The system, based on image processing, is developed using rocks sampled from different Calabrian quarries, some of which were used in historical times for the construction of artifacts of historical and archaeological interest [8,9,10].

The main contributions of this work are listed in the following:The paper proposes a methodology to be used in the stone recognition context of the main Calabrian quarries that, to the best of our knowledge, represents the first attempt in the stone literature;The paper proposes to use in the context of stone classification a Two-Stage Hybrid Model that joins the DL approaches with ML algorithms;The paper shows a set of experiments by which it is possible to take out some considerations on the best combination of DL plus ML techniques to be used in the stone recognition task.

The remainder of the paper is organized as follows. After a review of related literature (Section 2), we give a description of the materials used in our research (Section 3). A brief description of pre-trained CNN models and classification methods are provided in Section 4. Section 5 provides an introduction of the Two-Stage Hybrid Model composed of a CNN network followed by a traditional ML algorithm. Section 6 describes the experiments to evaluate the performance of the provided Two-Stage Hybrid Model showing the achieved results. Section 7 concludes the paper with some final considerations.

## 2. Related Work

In the last few years, many researchers focused their studies on the DL approach for many different tasks. In particular, the attention has been concentrated on the CNN that represents an important technique able to resolve many different issues regarding different aspects such as speech recognition, natural language processing, bioinformatics, and image classification [11]. Our attention is focused on image recognition issues and, in particular, our application domain regards stone recognition. Many different works exist in literature about stone classification through image processing and many works exist on neural network and DL approaches applied to this domain. In the remainder of this section, we show the main works in order to contextualize our research.

### 2.1. *Convolutional Neural Network* (CNN) for Classification

Many papers have faced the topic of image processing and classification using DL and *Convolutional Neural Network* (CNN) solutions. In [12], an evaluation of an image classifier using traditional computer vision and DL approaches is provided. They use an Inception-V3 architecture and their own CNN called TinyNet built from scratch. The accuracy and loss attributes are provided as a result of the evaluation. In [13] the use of DL approach for image classification is provided. The authors analyzed and implemented a VGG-16 model for performing image classification into different categories. Moreover, they provide a methodology for more accurate classification of images. In [14] and in [15] the authors show the use of the CNN approach for visual object recognition using only SVM, in the first case, and Softmax and SVM classifiers, in the second one. Moreover, the authors of the second paper demonstrate a small but consistent advantage of replacing the Softmax layer with a linear SVM. A work based on pedestrians using CNN and SVM techniques is proposed in [16]. Their tests show that the proposal is able to quickly and reliably detect the pedestrian targets on the Caltech data set. In [17] the authors propose an image classification model applied for identifying the display of the online advertisement using a *Convolutional Neural Network* (CNN). The proposed CNN considers two parameters (n,m) where *n* is a number of layers and *m* is the number of filters in convolutional layers that are chosen on the basis of a series of experiments that they present in the paper. In [18] the authors investigate the use of a deep convolutional neural network CNN for scene classification. They experiment with two simple and effective strategies to extract CNN features, first using pre-trained CNN models as universal feature extractors, and then, domain-specifically fine-tuning pre-trained CNN models on their scene classification dataset. In [19] the authors propose a CNN architecture using the MNIST handwritten dataset in order to validate it. They utilize an optimized hardware architecture with reduced arithmetic operations and faster computations implemented on an FPGA accelerator. Another paper focusing on computational architecture is [20]. The authors implement an image classification CNN using a multi-thread GPU on the CIFAR10 dataset. In [21] the authors deal with the problem of synthetic aperture radar (SAR) image classification. They design a deep CNN architecture proposing a microarchitecture called Compress Unit (CU). Their architecture, compared with other networks for SAR classification in literature, results in being more performed and efficient. Other works exist that compare classification approaches in order to show the best choice for their applicative domain. An investigation on supervised classification is in [22] where the authors evaluate the performances of two classifiers as well as two feature extraction techniques: Linear SVM and Quadratic SVM. An exploration of the hybrid CNN solution for image classification is provided in [23] where the authors provide a comparative study of seven CNN-based hybrid image classification techniques showing the results in terms of their accuracy. A specific topic of butterfly recognition is studied in [24]. The power of DL approaches has shown the capability of the CNN of discovering with accurate results the different varieties of these insects. They propose two CNN approaches building from scratch their neural model able to classify butterfly images. A problem of plant classification is analyzed in [25] through the use of two different hybrid CNN models implemented by the authors from scratch. They used three different datasets, namely LeafSnap, Flavia, and MalayaKew Dataset utilizing the data augmentation approach for better performing the training phase. Their study shows good results for the proposed models in terms of accuracy.

### 2.2. Stone Classification

A lot of works exist on this topic in literature. Many researchers face the stone classification issue taking into account many different approaches that involve earth science and the mining industry. In [26] the authors have presented some possible approaches to the development of an expert system for the automatic classification of granite tiles. Based on recent results on color texture analysis, they have proposed a set of visual descriptors which provide good classification accuracy with a limited number of features. In [27] the authors investigate the problem of choosing adequate color representation for automated surface grading. Moreover, they discuss the pros and cons of different color spaces basing their study on a dataset of 25 classes of natural stone. In [28] the authors describe a methodology for a correct and automated granite identification and classification by processing spectral information captured by a spectrophotometer at various stages of processing using functional ML techniques. In [29] the authors describe an approach for texture classification on a dateset of different stones. They have worked on extracting statistical features from histogram of grain components. So, they have provided a computable feature vector which has most meaningful information of texture. In [30] a novel approach to rotation and scale invariant texture classification is introduced. The proposed approach is based on Gabor filters that have the capability to collapse the filter responses according to the scale and orientation of the textures. Their experiments have shown the goodness of the proposed approach compared with other methods existing in the literature. In [31] the authors deal with the texture classification issues. In this paper, the authors propose an approach that uses both the Gabor wavelet and the curvelet transforms on the transferred regular shapes of the image regions. They show some experiments on texture classification demonstrating the effectiveness of the proposed approach.

A computer-vision-based methodology for the purpose of gemstone classification on 68 different classes of gemstones is provided in [32]. The authors utilize a series of feature extraction techniques used in combination with different ML algorithms. Moreover, they also use a DL classification with two ResNet models: ResNet18 and ResNet50. They provide results of classification methods against three expert gemmologists with at least 5 years of experience in gemstone identification showing the difference in time response between human and automatic approaches.

Other literature works that use the DL approach for automatic stone classification is [33], where automatic recognition and classification of granite tiles is the object of study using CNN networks such as AlexNet and VGGNet for a fine-tuning pre-trained approach, or [34] where the authors implement a classification model of ornamental rocks through the analysis and classification of images, using machine learning algorithms.

The use of the *Transfer Learning* (TL) approach for mineral microscopic image classification is reported in [35]. The authors show the system behavior using four mineral image features extracted by an Inception-V3 CNN network. Moreover, the features extracted are used for classification purposes throughout different ML methods such as: *Logistic Regression* (LR), *Support Vector Machine* (SVM), *Random Forest* (RF), *k-Nearest Neighbors* (kNN), *Multilayer Perceptron* (MLP), and GNB. As a result, they found that LR, SVM, and MLP have a significant performance among all the models, with accuracy of about 90.0%. This last contribution, which is one of the literature works used for conceiving our idea, is of proposing a hybrid model composed of two stages based on DL and ML approaches: the first one used for feature extraction and the second one used for performing stone classification. So, on the basis of these literature works we have proposed a methodology and a model to be used in the context of stone recognition proposing the joining use of four different CNNs and five different ML classification algorithms, also showing the best combination to be used.

### 2.3. Main Paper Contributions

This literature review presents the scientific community effort in this research field, also showing how the new AI approaches are largely used in the context of stone classification. From this study, it emerges that many researchers propose DL- or ML-based approaches but our Two-Stage Hybrid Model is distinguished for the provided methodology/modeling and represents a good solution for image recognition. Many studies deal with stone recognition using classical approaches based on texture and color space that represent very complex and resource consuming techniques. Other works introduce approaches based on AI techniques, but no one considers more CNNs (four CNN models) combined with different classifier algorithms. So, on the basis of these studies, in this work we propose a system model to be used in stone classification based on a hybrid approach that consists of a two-stage model in which, in the first stage, we apply the use of the DL approach based on four different CNN networks and, in the second stage we propose the use of ML techniques in order to perform image classification. Our study proposes a methodology and a modeling that can be used in different contexts of stone classification. Moreover, it uses the well-known TL approach in the first stage, in order to take advantage of feature extraction based on a large image database as ImageNet, passing this information to the second stage that, based on ML algorithms, performs the classification. The TL approach permits the avoidance of creating a CNN from scratch, making the project less complex and onerous in terms of time and resources.

So, in the following, the main contributions of this work are listed:Stone recognition of the main Calabrian quarries that, to the best of our knowledge, represents the first attempt in the stone literature;Two-Stage Hybrid Model proposal able to join the DL approaches with ML algorithms;Methodology for stone classification purpose giving indications to face with this specific task;Experimental tests for providing the best combination of DL and ML techniques to be used in the stone recognition task.

## 3. Materials

If we compare the quarries of stone materials currently exploited in Calabria with those known until the early 1900s and reported by [9], we find that today at least 70% of the historical quarries in Calabria are no longer exploited. Moreover, most of them have totally lost their historical knowledge and exact location. Other studies [10,36], recently conducted by the Calabrian Superintendence, show evidence of ancient quarries, located mostly on the coastal areas of Calabria, dating back to the Hellenistic and Roman period. This shows that Calabria, since ancient times, has been for many civilizations the place of preferential supply of stone materials used to realize artistic artifacts and ancient architectural buildings.

An easy-to-use tool, capable of identifying the quarries with which an ancient stone artifact was made, would make an important contribution to historical knowledge of trade relations between peoples of the same period. For this reason, it was decided to work on the most representative stone materials of the Calabria Region (Southern Italy), from the five provinces of Calabria (provinces of Reggio Calabria, Vibo Valentia, Catanzaro, Cosenza and Crotone). The location of the quarries is shown in Figure 1.

### 3.1. Stone Materials in Calabrian Provinces

The studied stone materials come from 25 quarries; 10 samples, representative of the geological outcrops, were sampled for each quarry. Figure 2 shows the 25 types of stone materials used for the classification in our Two-Stage Hybrid Model. Table 1 shows the historical name of the stone, the city in which the quarry is located and the geological classification of the stone. The studied rocks include magmatic rocks such as granodiorites, diorites and porphyrites, sedimentary rocks, such as sandstones, calcarenites and limestones, but also metamorphic rocks, such as marbles, schists, metabasites and serpentinites (Figure 2 and Table 1).

### 3.2. Image Acquisition System

To acquire the images, the stone samples coming from each Calabrian quarry have been cut through a petrographic cutter machine in order to obtain perfectly flat and smooth surfaces. The flat surface obtained, for all 250 samples, was acquired in three different modes, using two simple tools: a smartphone and a flatbed scanner.

1.The first typology of images was acquired using a smartphone Samsung Galaxy Note 4, with 16 Mpixel camera and a resolution of 4608 × 3456 pixels. The acquisition was performed under standard conditions, illuminating the sample with an LED illuminator, inserting the flash of the smartphone and always keeping constant the distance between the smartphone and the sample surface (10 cm).2.The second typology of images was acquired by flatbed scanner, with reflected light, using an Epson Perfection 2400 Photo scanner, with a resolution of 600 dpi (image type: 24-bit colors). During the acquisition, all the filters were removed and the samples were carefully covered with a synthetic black and thermal cloth to normalize the acquisition and to perform it in standard condition.3.The third typology of images was acquired using the same flatbed scanner and the same conditions of the previous typology, the only difference is that the acquisition was made on the wet surface of the samples, in order to simulate a polished effect of the stone.

## 4. Pre-Trained CNN Models and Classification Algorithms

Object recognition is a key technology based on AI techniques. Recently, ML and DL techniques have become commonly used approaches to solve problems related to object recognition. In DL, a computer model learns to perform classification tasks directly from images. Recent developments have allowed DL to progress to such an extent that it surpasses humans in some activities, such as the classification of objects in images. There are two approaches to performing object recognition using DL:1.Train a model from scratch;2.Using a pre-trained DL model (*Transfer Learning* (TL) technique [1]).

One of the biggest advantages of the current DL approach is the ability to have access to pre-trained networks. In this way, it is possible to avoid having to spend many hours, if not days, training the network, and directly use the architecture of the network and the weights obtained from the training, downloading them from the Internet. It is one of the advantages offered by the “Open Source” approach adopted to a large extent. Another advantage is the capability of solving the problem of lacking a large training dataset. The features from multiple fully-connected layers are combined with different weights and used to train different algorithms for image classification.

Our image classification system is based on the TL approach [1], a process that consists of refining a previously trained model through a re-training of the specific images used for the recognition. In our study, we utilized four CNN models and the performance of these models was evaluated. All the pre-trained models were trained on the ImageNet dataset, and each model is briefly explained in the following sections.

In particular, we used the pre-trained model as a feature extractor. We know that a DL model is basically a grouping of interconnected layers of neurons, where the last one acts as a classifier [37]. By removing the final layer of the considered pre-trained CNN network, the output of the penultimate layer, representing the feature vector, can be used as input to the ML classifier in order to perform the stone recognition on the basis of our dataset and, then classify stone as belonging to one of the 25 different classes. In fact, in the pre-trained network, the last layer classifies on the basis of the large database called ImageNet on 1000 different classes of objects. So, the main purpose of the pre-trained CNN is to provide the feature vector extracted by the large image database and use it to perform classification on our stone database.

We have used, in our hybrid solution, four different types of classifiers in order to compare the performance of each one and indicate the most promising solution in solving an image classification task. The network code is open source and is provided for the Tensorflow framework [38].

### 4.1. ImageNet Dataset

ImageNet is a large image database, created for use in the field of computer vision and in the field of object recognition [39]. The dataset consists of more than 14 million images with the indication of the objects they represent. Identified objects have been classified into more than 20,000 categories: some categories of frequent objects, such as “balloon” or “strawberry”, consist of several hundred images [40]. Since 2010, a competition called the ImageNet Large Scale Visual Recognition Challenge (ILSVRC) is held every year. On this occasion, software programs are made to compete to classify and correctly detect objects and scenes contained in the images. As part of the competition, a reduced list of images with objects belonging to a thousand non-overlapping categories is used [41].

### 4.2. CNN Models

The *Convolutional Neural Network* (CNN) or ConvNet [42] is one of the most common algorithms for DL, a type of ML in which a computer model learns to perform classification tasks directly from images, video, text or sound. CNNs are particularly useful for finding patterns in images to recognize objects, faces and scenes. They learn directly from image data, using patterns to classify images and eliminating the need for manual feature extraction. CNNs offer an alternative approach that automates feature learning using large databases of samples, called training sets, which represent an application domain of interest. A CNN can have tens or hundreds of layers capable of learning to detect the different features of an image, see Figure 3.

On the basis of the literature works analysis and of the main CNN networks proposed and used by researchers to perform their experiments, the choice of the CNNs that we have proposed in our Two-Stage Hybrid Model fell back on the following neural networks: VGG-16, VGG-19, Inception-V3 and ResNet50. In this section, these four CNNs, used for our Two-Stage Hybrid Model, are briefly introduced.

VGG-16 is a neural network architecture designed by the Visual Geometry Group, the department of engineering sciences of the University of Oxford, with 13 convolutional layers and three fully connected layers for classification and detection tasks, as shown in Figure 4. It accepts, as input, images with a resolution of 224 × 224 pixels in RGB, has an output of 4096 features, and input of a classification layer.

The VGG-19 is a CNN with 19 layers among convolutional, pooling and fully connected layers trained on the ImageNet database. It has an architecture very similar to the previous VGG-16 version as it is possible to view in Figure 5 where the output of the network is a 4096 feature vector which is then used as input of a classification layer.

Inception-V3 was developed by Google and trained on the ImageNet database composed of 1000 different classes [43,44]. This model uses the inception modules which take several convolutional kernels of different sizes and stack their outputs along the depth dimension in order to capture features at different scales, see Figure 6.

ResNet won ILSVRC in 2015, the ImageNet Large Scale Visual Recognition Challenge [45]. Five different versions of ResNet exist, with a number of layers from 18 to 152 and with a consequent explosion of complexity. ResNet50 is the version with 50 layers, see Figure 7.

### 4.3. Classification Techniques

Once the features are extracted by the CNN network in the first stage of our Two-Stage Hybrid Model, these data are passed in input to the second stage where a set of ML classifiers are used for performing stone classification on the 25 different stone classes. The choice of these classifiers is, also in this case, due to the previous analysis of literature manuscripts where a lot of authors proposed mechanisms based on different ML algorithms. We have selected the most used and proposed different machine learning methods in order to perform our tests on the considered stones belonging to Calabrian quarries. A plethora of classifiers could be used but these five are used for the most part of the considered works. So, in this paper, the following ML classification methods are used:*Softmax* or *Multinomial Logistic Regression* (MLR) [2], representative of regression models;*Support Vector Machine* (SVM) [3,46], an example of linear models;*k-Nearest Neighbors* (kNN) [4], representative of density or instance based models;*Random Forest* (RF) [5,47], representative of ensemble models;*Gaussian Naive Bayes* (GNB) [6], an example of probabilistic models.

For more details on these classifier algorithms please refers to [48].

## 5. Two-Stage Hybrid Model

With the term *hybrid model* we mean an approach that makes use of a *Deep Learning* (DL) network together with a classical *Machine Learning* (ML) algorithm. The joined use of these two different approaches can give many advantages to the classification purpose.

The proposed hybrid model consists of two stages as it is possible to view in Figure 8. The first stage of the model guarantees an automatic feature extraction phase that is used as input for the second stage, which has the task of performing the classification phase.

Moreover, in order to make more efficient the phase of the feature extraction we have used in our Two-Stage Hybrid Model the well-known *Transfer Learning* (TL) approach that, throughout models of CNN networks pre-trained on a large image database, guarantees a more rapid feature extraction phase avoiding building the neural network from scratch. This approach, as proved by the literature, guarantees optimal results and allows for efficient creation, on the basis of a set of filters used in the CNN, a feature vector composed of numerical values representing the main information of the image in a very short time period.

The *Two-Stage Hybrid Model* that we propose in this work is shown in Figure 8. We use four different pre-trained CNNs from the main networks provided in the literature and briefly explained in Section 4.2 in the first stage of the model and five different ML algorithms in the second stage, see Section 4.3, in order to accomplish the image classification on our specific dataset. So, each CNN model has been used as a feature extractor for the five different classifiers. The main output parameters are reported in the next section in order to evaluate how each model performs providing a useful comparison of an original context represented by stones of Calabrian quarries.

## 6. Experiments: Results and Discussion

In this section, we give a detailed description of all the experiments we performed with our proposed two-stage architecture presented in the previous Section (Section 5). The heart of this project is to perform the right class prediction for the considered stones’ images. We have several images for the input, subdivided between *training dataset* (80% of the total dataset) and *test dataset* (20% of the total dataset). It might be interesting to see what the Two-Stage Hybrid Model guesses for classes of images it never saw during training.

In order to determine the performance of a neural network, it is important to take into account some characteristic parameters that can help to indicate the goodness of the approach. In the context of AI, the confusion matrix, also called the misclassification table, returns a representation of the accuracy of statistical classification. Each column of the matrix represents the predicted values, while each row represents the real values. The element on row *i*-th and column *j*-th is the number of cases in which the classifier has classified the “true” class *i*-th as class *j*-th. Through this matrix, it is observable if there is “confusion” in the classification of different classes. The confusion matrix provides a lot of information. However, more concise metrics are often useful, such as: accuracy, precision, recall and F1-score.

### 6.1. Experimental Environment

In order to perform classification experiments, a workstation equipped with an Intel i9 10900K CPU with 32 GB RAM DDR4, an Nvidia 3060Ti graphic card and a 512 GB SSD and a Python 3.8.10 was used. Moreover, we have installed some additional libraries such as Matplotlib, Pandas, Tensorflow, and NumPi in order to analyze and perform classification on our dataset.

### 6.2. Our Dataset

In this study, the image classes, that represent the different object categories, are the 25 different stone types of Calabrian quarries. We need to ensure that the images have the right size for each considered CNN. In particular, a resolution of 224 × 224 × 3 is used for VGG-16, VGG-19 and ResNet50 models and a resolution of 299 × 299 × 3 for the input of Inception-V3 one. So, a little pre-processing operation was made on the input image dataset in order to match the specific image resolution requirements of the considered CNN architecture.

### 6.3. Augmentation of Our Dataset

In order to make our experiments with a sufficient number of images, we have increased our dataset by creating new images through a simple elaboration of our data samples by performing the so-called *Data Augmentation* technique [49].

In order to augment the training dataset for our experiment and consider the large image resolution of our dataset, we have manipulated our input images in a very simple way. We have cropped the original image in five different parts both in horizontal and vertical axes creating so 25 different images from the original one as it is possible to view in Figure 9. This has allowed us to increase the number of images by a 25 factor, creating a consistent dataset of stone images.

### 6.4. Classification Results

As described in Section 3, for our experiments, images from three different typologies of acquisition have been used. For each type of stone, 10 different images with each type of acquisition technique have been created. Then, for each type of stone, such as ASL, CAG, CAP, etc., we have used 10 × 3 = 30 different images on which we have performed the technique of data augmentation, as previously described in Section 6.3, in order to increase the input dataset both for training and test campaigns. As we said in the previous section, we extracted 25 different images from each one, then, we obtained 25 × 30 = 750 different images for each stone typology. From these images, 600 were used for conducting the training of the neural network and 150 were used for performing the test campaigns in order to evaluate the goodness of the network. This corresponds to having, respectively, a training set and a test set composed of 80% and 20% of the overall dataset.

Different experiments were conducted using the image database created by the stones acquisition process. We have performed experiments on every single typology of stone acquisition, that is, using first only the database comes from smartphone acquisition, second, the database comes from the acquisition with the flatbed scanner with reflected light and, finally, the database created using the flatbed scanner with reflected light on the wet surface of the samples. These experiments have shown the advantages of the automatic classification through the CNN approach using the different *Classification Algorithm* (CLFs)

The total number of CNN parameters is reported in Figure 10 (left) in order to show how complex is the CNN network and provide a comparison between the four CNN models used in our experimentation. How it is possible to view in Figure 10 (left), the Inception-V3 and ResNet50 CNN models have a lower number of parameters. Moreover, Figure 10 (right) reports the inference time comparison between the four CNN models, both to infer on training and on test samples. How it is possible to note that the ResNet50 model, due to its small parameter number, is able to infer in a shorter amount of time in both sample sets.

In Figure 11 we show the confusion matrix of the first two used classifiers, Softmax and SVM, with the ResNet50 CNN model that results in the best CNN to be used together with the classifiers in order to have the best results in terms of accuracy. In this section, only the confusion matrices of the ResNet50+CLF hybrid model are reported. It is possible to observe that both approaches reached similar results on the test set used for experiments. The network, both with Softmax and SVM classifier, is able to perform prediction with high accuracy. It is possible to observe that the classes that present some recognition problems in all Two-Stage Hybrid Models are those related to the family of granite stones, that is GRB, GRD, GRS1, GRS2 and GRS3 families. This is due to a very similar texture of these types of stones that also makes it difficult for an expert eye to capture the differences.

In a similar way, we have performed experiments using *k-Nearest Neighbors* (kNN) and *Random Forest* (RF) classifiers. In the following, the confusion matrices are shown for both classification algorithms. As it is possible to observe in Figure 12, also in this case, the recognition that presents more issues regards the granite stone classification. It is possible to view that these two types of classifiers have similar accuracy to the first two considered in the previous experiments. This means that Softmax and SVM have similar performance to kNN and RF algorithms.

The last used classification algorithm is the GNB and, in Figure 13 the confusion matrix extracted by image recognition tests is reported. As shown in the figure, this last method presents the worst results in terms of classification accuracy.

The accuracy that we have calculated for all the conducted experiments is shown in Table 2 and Figure 14 (left). Then, the experiments showed that these algorithms have optimum performance in pattern recognition purposes: all algorithms are able to recognize the most part of the input images but, between all, the Softmax, SVM, RF and kNN approaches allow to reach very high accuracy values representing the best candidates for image classification in this applicative domain.

Other than the accuracy, in this section, we also show other metrics: precision, recall and F1-score in order to prove the goodness of the classification. With precision metric, the system shows how on the true classification the most part is correct; with the recall instead, the system is able to cover the most part of the true positive; F1-score gives a joined metrics between precision and recall. Figure 14 and Figure 15 provide the metric values for each Two-Stage Hybrid Model considered in our tests. As it is possible to observe, the two-stage hybrid approach provides a very good performance in almost all tests reaching optimal results using a ResNet50 CNN in the first stage and a kNN ML algorithm in the second stage of the hybrid model.

So, after this set of experiments that have used the Two-Stage Hybrid Model combining each DL technique with each ML algorithm, it is possible to make some considerations in order to conclude our work. The proposed hybrid model consists of a two-stage approach that makes use of DL techniques in the first stage with the task of performing feature extraction and ML algorithms in the second stage with the task of performing classification and, then operating the stone recognition so as to attribute the right class to the specific stone. The use of the combination of DL and ML approaches resulted in a very high-performing system able to recognize the belonging class very well. The most accurate combination that emerged from the results was based on the ResNet50 CNN model in joining with the kNN classifier (ResNet50 + kNN). This combination is able to guarantee a high accuracy in the stone recognition as proved by Table 2 and Figure 14 and Figure 15. The ResNet50 network also resulted in the best in terms of the number of CNN parameters and inference time as reported in Figure 10.

## 7. Conclusions

In this paper, an automatic stones classification approach based on a two-stage hybrid architecture able to classify different stone classes in the Calabria area (Southern Italy) is presented. The obtained results are pretty impressive. The neural network models are able to reach amazing results in the prediction process on the input images provided to the network. The proposed Two-Stage Hybrid Model based on DL and ML techniques results in being a more promising approach for stone recognition issues. From the conducted analysis, it emerged that the only classes that present some minor issues are those related to granite typologies that result quite complex also for a careful eye of an expert in this field. However, this two-stage hybrid approach, which uses *Deep Learning* (DL) CNN models together with different *Machine Learning* (ML) *Classification Algorithm* (CLFs), permits to create a system that is very powerful and able to reach optimal performance in terms of image recognition. It exploits the power of DL for the phase of feature extraction, which represents the more complex phase, and leverages the classical ML algorithms to perform the classification phase. Moreover, in order to avoid creating the CNNs from scratch, the proposed Two-Stage Hybrid Model is based on the *Transfer Learning* (TL) paradigm that is able to exploit pre-trained networks on large datasets such as ImageNet for reducing the phase of feature extraction. In fact, the CNN in the TL mode is able to infer on both training and test sets in a very quick manner as shown by provided results. The most promising combination that emerged from tests is based on the ResNet50 CNN model together with a kNN classifier. It guarantees high accuracy and allows us to obtain the best results in terms of CNN parameter number and inference time. Furthermore, this type of approach shows that there is a concrete possibility to build tools that are easy to use even for people who do not have geological knowledge; the applications could be numerous and range from the field of archaeometry and diagnostics, up to applications of automatic recognition in the field of the materials sciences.

## Figures and Tables

**Figure 1 sensors-22-06292-f001:**
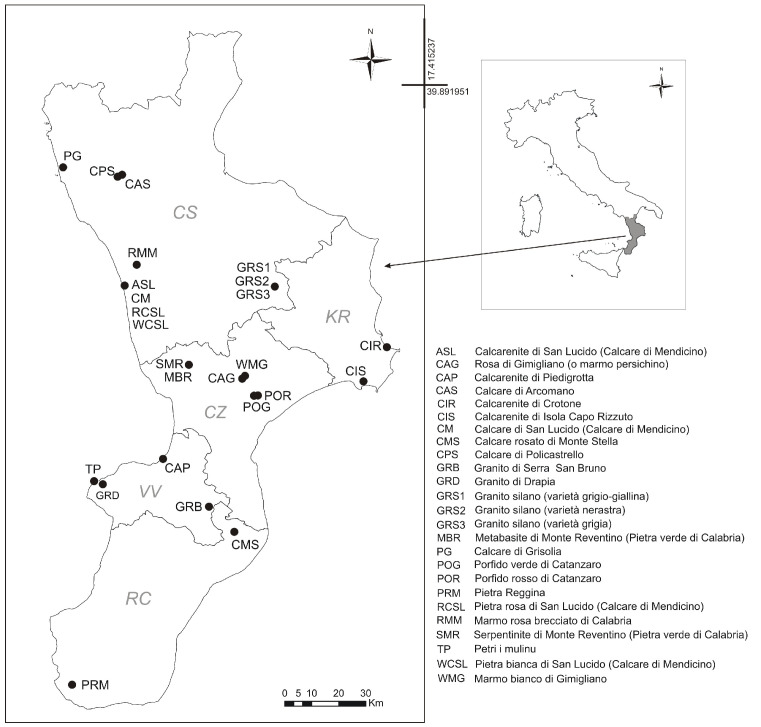
Location of the studied quarries in Calabria Region (Southern Italy). The legend shows the historical name of the stone materials studied.

**Figure 2 sensors-22-06292-f002:**
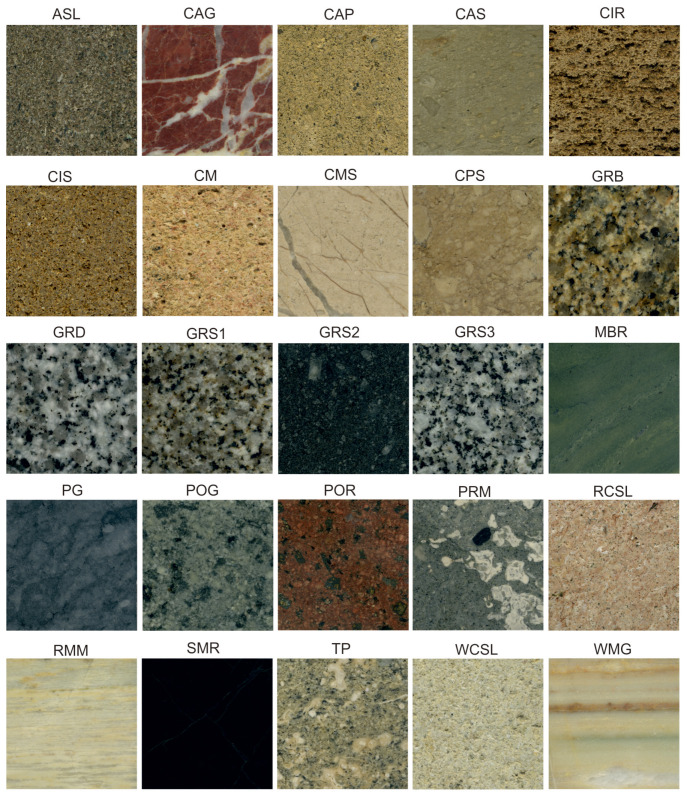
Macroscopic photos of the studied stone materials representative of each quarry. The photos were collected in reflected light using a flatbed scanner. The sizes of each photo are 5 cm × 5 cm.

**Figure 3 sensors-22-06292-f003:**
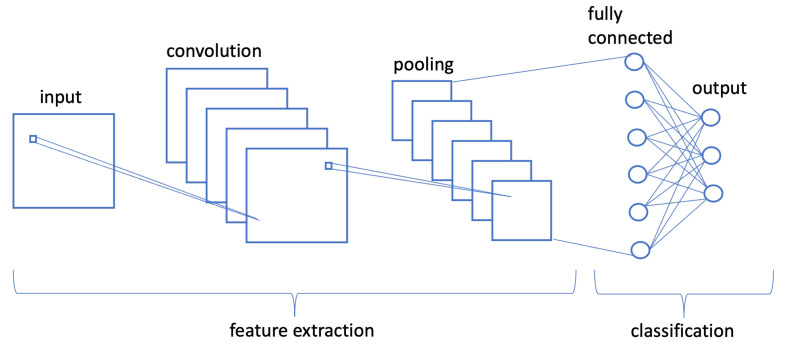
Example of CNN architecture.

**Figure 4 sensors-22-06292-f004:**
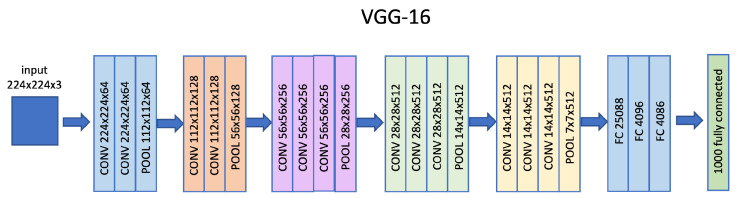
VGG-16 architectural model.

**Figure 5 sensors-22-06292-f005:**
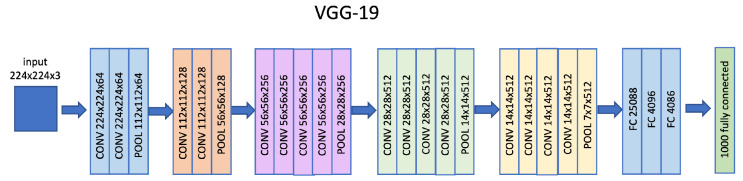
VGG-19 architectural model.

**Figure 6 sensors-22-06292-f006:**
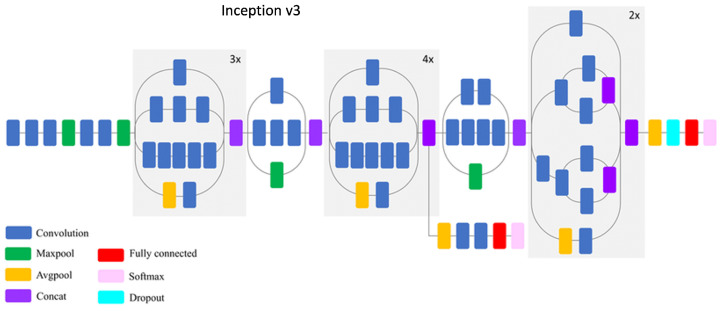
Inception-V3 architectural model.

**Figure 7 sensors-22-06292-f007:**
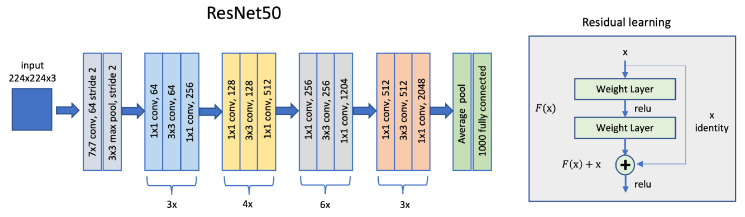
ResNet50 architectural model.

**Figure 8 sensors-22-06292-f008:**
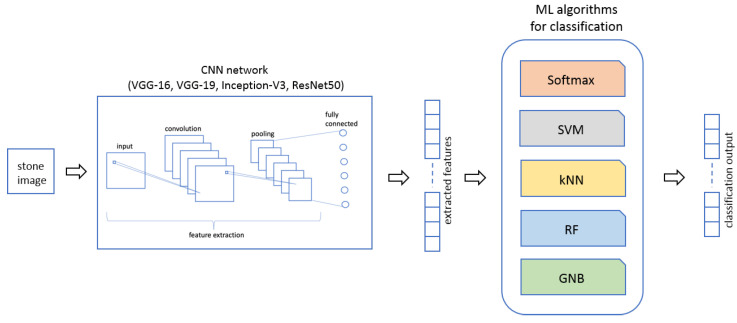
Two-Stage Hybrid Model used for stone classification.

**Figure 9 sensors-22-06292-f009:**
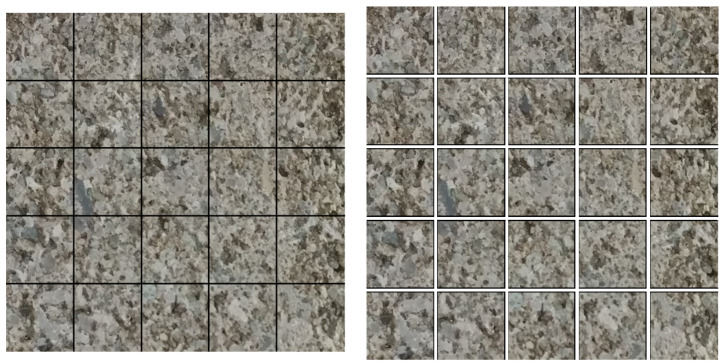
Example of data augmentation used in our experiments.

**Figure 10 sensors-22-06292-f010:**
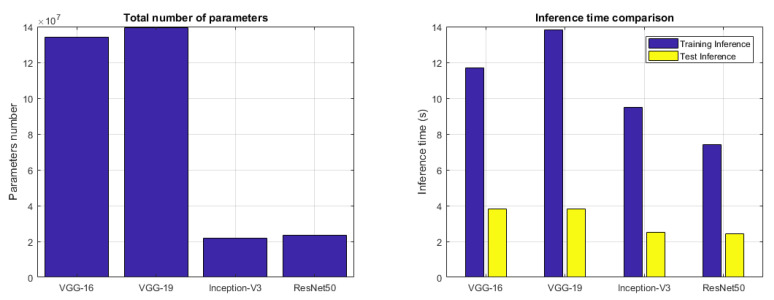
Total number of CNN parameters (**left**) and Inference time (seconds) (**right**) of each CNN model used in the hybrid architecture.

**Figure 11 sensors-22-06292-f011:**
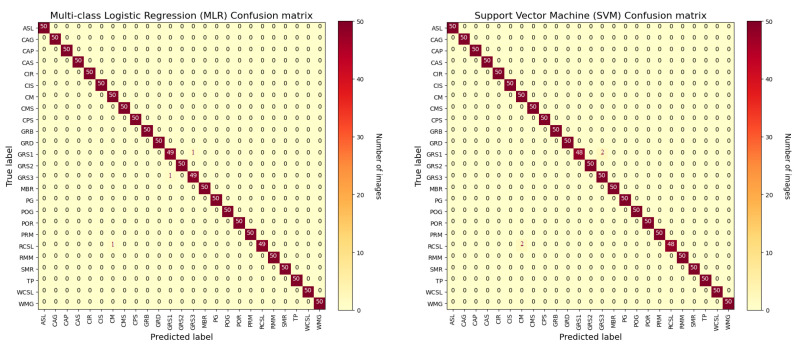
Confusion matrix for (**left**) Softmax (MLR) and (**right**) SVM classifiers with ResNet50 CNN model.

**Figure 12 sensors-22-06292-f012:**
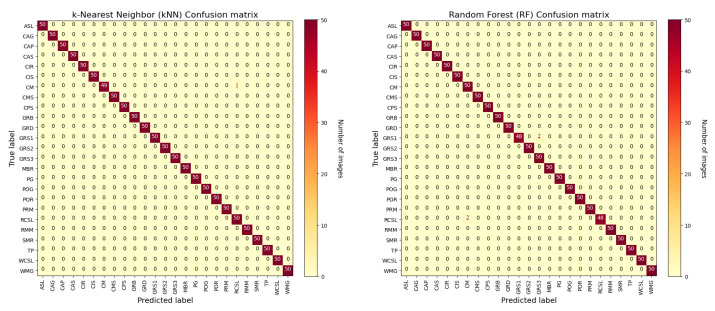
Confusion matrix for (**left**) kNN and (**right**) RF classifiers with ResNet50 CNN model.

**Figure 13 sensors-22-06292-f013:**
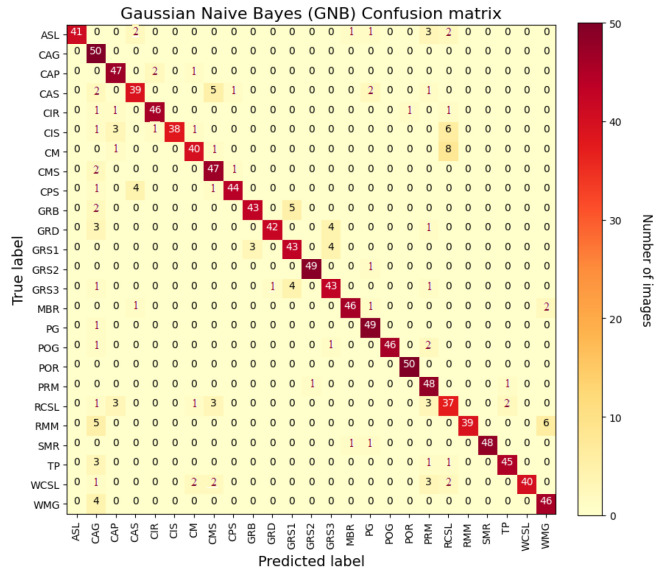
Confusion matrix for GNB classifier with ResNet50 CNN model.

**Figure 14 sensors-22-06292-f014:**
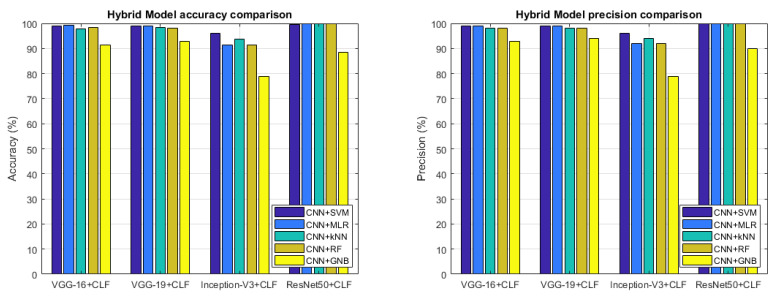
Accuracy (**left**) and precision (**right**) comparison.

**Figure 15 sensors-22-06292-f015:**
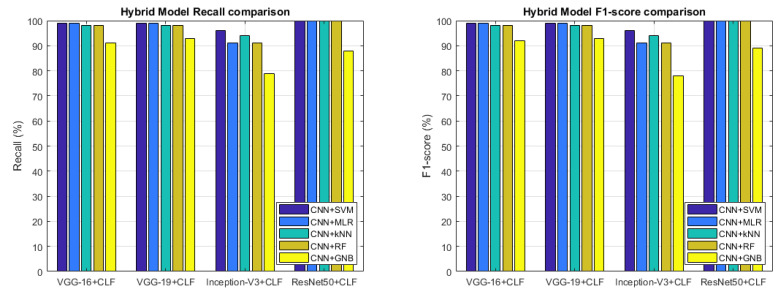
Recall (**left**) and F1-score (**right**) comparison.

**Table 1 sensors-22-06292-t001:** List of the stone materials studied from the five Calabrian provinces (Southern Italy).

Short Code of the Quarry	Historic Name of the Stone	Name of the City Where the Quarry Is Located	Geological Classification of the Stone
ASL	Calcarenite di San Lucido (Calcare di Mendicino)	San Lucido (Cosenza)	Calcarenite
CAG	Rosa di Gimigliano (o marmo persichino)	Gimigliano (Catanzaro)	Dolomitic Limestone
CAP	Calcarenite di Piedigrotta	Pizzo Calabro (Vibo Valenzia)	Calcarenite
CAS	Calcare di Arcomano	San Donato di Ninea (Cosenza)	Limestone
CIR	Calcarenite di Crotone	Crotone (Crotone)	Biocalcarenite
CIS	Calcarenite di Isola Capo Rizzuto	Isola Capo Rizzuto (Crotone)	Calcarenite
CM	Calcare di San Lucido (Calcare di Mendicino)	San Lucido (Cosenza)	Variable from limestone to dolomitic limestone
CMS	Calcare rosato di Monte Stella	Pazzano (Reggio Calabria)	Oolitic limestone (oosparite)
CPS	Calcare di Policastrello	San Donato di Ninea (Cosenza)	Evaporitic limestone
GRB	Granito di Serra San Bruno	Serra San Bruno (Vibo Valentia)	Granodiorite
GRD	Granito di Drapia	Drapia (Vibo Valentia)	Granodiorite
GRS1	Granito silano (varietà grigio-giallina)	San Giovanni in Fiore (Cosenza)	Granodiorite
GRS2	Granito silano (varietà nerastra)	San Giovanni in Fiore (Cosenza)	Diorite
GRS3	Granito silano (varietà grigia)	San Giovanni in Fiore (Cosenza)	Granodiorite
MBR	Metabasite di Monte Reventino (Pietra verde di Calabria)	Platania (Catanzaro)	Metabasite o greenschist
PG	Calcare di Grisolia	Grisolia (Cosenza)	Limestone
POG	Porfido verde di Catanzaro	Catanzaro (Catanzaro)	Dioritic green porphyry
POR	Porfido rosso di Catanzaro	Catanzaro (Catanzaro)	Monzonitic red porphyry
PRM	Pietra Reggina	Motta San Giovanni (Reggio Calabria)	Calcarenite
RCSL	Pietra rosa di San Lucido (Calcare di Mendicino)	San Lucido (Cosenza)	Variable from limestone or dolomitic limestone to calcarenite
RMM	Marmo rosa brecciato di Calabria	Montalto Uffugo (Cosenza)	Fine marble
SMR	Serpentinite di Monte Reventino (Pietra verde di Calabria)	Platania (Catanzaro)	Serpentinite
TP	Petri i mulinu	Tropea (Vibo Valentia)	Calcarenite
WCSL	Pietra bianca di San Lucido (Calcare di Mendicino)	San Lucido (Cosenza)	Biocalcarenite
WMG	Marmo bianco di Gimigliano	Gimigliano (Catanzaro)	Calce-schist

**Table 2 sensors-22-06292-t002:** Accuracy of CNNs plus CLFs.

	MLR	SVM	kNN	RF	GNB
VGG-16 (%)	99.0	99.3	97.8	98.3	91.4
VGG-19 (%)	99.0	99.1	98.4	98.2	93.0
Inception-V3 (%)	96.0	91.4	93.8	91.4	78.9
ResNet50 (%)	99.7	99.8	99.9	99.7	88.5

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
