# Peer review of "Automatic Stones Classification through a CNN-Based Approach"

_sensors, 2022, doi:10.3390/s22166292_

Round 1
Reviewer 1 Report
This paper presents an automatic recognition system for classifying stones belonging to different calabrian quarries.
The suggested approach is a hybrid one based on Deep Learning (DL) and Machine Learning (ML). The study topic (the stones used) is rather unique
and the study is comprehensive. The paper contains some merits. However, it requires major revision for possible publication. Here are some comments:
1) The main contributions are not clear since similar approaches have been proposed, as in the end of section 2 and references [29] - [32]. The authors
need to clearly indicate what is the main difference between the current paper and the above references, i.e., what is new in this paper.
2) The paper spends too many pages to describe existing and well known facts, such as Sections 4, 5, 6, which takes about half of the paper. For example,
the difference between VGC-16 and VGC-19 is very small and there is no need to repeatedly introduce both. The same goes to well known classification methods presented in
Section 6.
3) The proposed hybrid model is illustrated in Fig. 14. But from the results presented in the Exepriments section, it is not clear how this model will work
in practice. It seems the authors simply compare the results by using one of the CNN models to combine with one of the classification method and just
evaluate the results from different combinations. The readers would like to know the final and recommended model to be used in the end, particularly in
testing new given data (stone images in this area). But this is not clearly presented.
4) Since there are some related work in this area, [29] - [32], can the authors compared their method(s) with at least some of the existing works?
Finally, there are also some minor English errors/improvements. Some are listed below and the authors could find more:
a) In the Abstract: "an hybrid approach".
b) Line 127: Many researchers face ...
c) 8.4 Classification Report
d) Line 682: It exploit the powerful of DL (two errors here).
Reviewer 2 Report
1. 'Provenance' might not be used as a keyword
2. Some typos in the paper, e.g.
The development of technology and the exponential improvement of computational sciences has made it possible to create computer learning software.
Many different works exist in literature about stones classification through image processing and many works exist on neural network and DL approaches applied to this domain [10].
The accuracy and loss attributes are provided as result of the evaluation.
The try also to provide a methodology for approaching to an accurate image classification problem.
It exploit the powerful of DL .....
3. Please also check the grammar carefully.
4. Two papers related to the work of CNN and SVM for classification, thje authors may introduce this paper so as to broaden the scope of this paper.
4.1 Fengyu Gao, Jer-Guang Hsieh and Jyh-Horng Jeng, “A Study on Combined CNN-SVM Model for Visual Object Recognition”, Journal of Information Hiding and Multimedia Signal Processing, Vol. 10, No. 4, pp. 479-487, December 2019
4.2 Yong-qiang He, Qin Qin and Vychodil Josef, “A Pedestrian Detection Method Using SVM and CNN Multistage Classification”, Journal of Information Hiding and Multimedia Signal Processing, Vol. 9, No. 1, pp. 51-60, January 2018
Reviewer 3 Report
The authors present a hybrid DL- and ML-based approach to classify stones from Calabrian quarries. The feature extraction component is addressed using CNN, whereas traditional ML methods are used to perform the final classification (SVM, etc.) in a hybrid approach. A transfer learning approach is used for training. The authors describe the image acquisition procedures to build the training data set, and they compare the performance of several combined CNN architectures and ML classification methods on stone images from Calabrian quarries.
Overall, the only clear paper novelty seems to be in the application of hybrid CNN+ML classification tools to the automated classification of stone images. Unfortunately, the paper clarity of presentation needs major improvements, which will hopefully lead to the identification of additional (computational?) novelties, if any. Besides the English style, I noticed that the authors go to great lengths to describe well-known techniques, databases and metrics. In my opinion, shortening those lengthy descriptions might help the reader focus more on the novelties/results. Once the work is more clearly presented, I suggest the author explicitly list the novelty (or novelties) of the work in the introduction, and then follow-up on them in the remainder of the papers.
Based on the above, in my opinion the paper might become suitable for publication after major revisions. I am listing below my comments to be addressed before reconsidering the paper for publication.
· Please discuss somewhere in the paper the shortcomings of the (best?) stone classification techniques covered in Section 2.2, and the novelties of your approach with respect to them.
· The authors state (from line 522) that “In fact, the first layer of the model guarantees an automatic feature extraction phase that is used as input for the second layer that has the task of performing the classification phase.” The authors also state in lines 322-323: “This output from the penultimate layer is then passed to a further set of layers, followed by a classification one”.
Elsewhere in the paper (lines 317-319), they state that DL networks consist of a final classification layer (“We know that a DL model is basically a grouping of interconnected layers of neurons, with the last one acting as a classifier [36].”).
What is the difference between their approach and traditional DL networks? Can they elaborate more about the proposed modification? What is the role of the additional set of layers, and what kind of features are output from the DL? How did they choose the modification, and have they tried different ones? Please discuss why the proposed approach is different from any traditional DL network with a classification layer at its end.
· The authors state (from line 524) that “Moreover, the hybrid model can be used together with a Transfer Learning (TL) approach that, using models of CNN networks pre-trained on large image database, guaranteeing a more rapid feature extraction phase avoiding to build from scratch the neural network. This approach, as proved by literature, guarantees optimal results.” Is the use of TL enabled in any way by using the specific proposed hybrid model? If so, please explain. If not, please rephrase the sentence and make it clear that the TL is not a novelty of the paper, but a well-known approach you use to get more efficient feature extraction.
· Please explain the data augmentation procedure you employed more in detail. It is not clear if you applied rotations, translations, zooming and more, or just cropping. Figure 15 seems to suggest just cropping, but the text below from Section 8.3 seems to be incongruous.
o “These alterations include a range of specific operations to make on the images that have the task to perform changes and manipulations such as flips, crops, translations, rotations, zooms and much more.”
o “In order to augment the training dataset for our experiment and considering the large image resolution of our dataset, we have manipulate our input images in a very simple way. We have cropped the original image in five parts both in horizontal and vertical axes creating so 25 different images from the original one as it is possible to view in Figure 15.”
· In my opinion the paper has way too much information about well-known techniques and datasets (DL, CNN, ImageNet and other datasets or architectures, classification techniques and metrics, for instance lines 230-245, 274-302, Sections 5.1 to 5.3, Sections 6.1 to 6.5, Section 8.4). These lengthy descriptions can be significantly shortened, for instance by just citing relevant literature. That way, the authors can focus more on explaining details and novelties of their work, as suggested in the above items.
· In the literature review, the authors state that “Many different works exist in literature about stones classification through image processing and many works exist on neural network and DL approaches applied to this domain [10].” However, it seems to me reference [10] is about image classification in general, not only with application to stone classification. Please modify the sentence if you agree, or explain why your citation is correct.
· Figure 7 is not clear. The authors state that “In particular, the inception modules are indicated with A, B and C and are based on convolution and pooling layers, where “n” indicates a convolution layer and “m” indicates a pooling layer. The parameters n and m are the convolution dimensions.” Where is “m” in the figure?
· In line 260-261, you state “The increase in data is a way to reduce the previously described phenomenon of overfitting.” It does not seem you have discussed it previously in the paper. Please modify the sentence, or add a discussion about overfitting where appropriate.
· Can you improve the coloring scheme for Figg. 17 to 19? It is difficult to appreciate differences in off-diagonal elements, which are important to visualize the difference in performance between the various hybrid models.
· The authors state that “ResNet50 model, despite its small parameters number, is able to infer in a minor amount of time in both sample sets.” Can they explain why that is surprising to them? To my knowledge, inference speed is not necessarily inversely proportional to parameter number.
· As I mentioned, in my opinion the paper needs a major revision of the English style to make it more easily readable. To this end, I am listing a non-exhaustive list of typos/style improvements and suggestions for the authors’ reference.
o Line 20: please change “we have developed complex skills” to “humans have developed complex skills”
o Line 23: Please change “a certain pattern, that can be” to “a certain pattern, which can be”
o Line 27: please change “depending on the application they are performing.” to “depending on the application.”
o Line 32: please change “measures” to “metrics”
o Line 35: please change “classification of the input in” to “classification of the input into”
o Line 37: please change “Convolutional Neural Network (CNN) network” to “Convolutional Neural Network (CNN)”
o Line 37: please change “for performing the feature extraction” to “to perform the feature extraction”
o Line 38: please change “one of the most important step” to “one of the most important steps”
o Lines 38-39: please change “for avoiding of creating” to “to avoid creating”
o The sentence about TL is in the introduction is not clear (lines 39-40): “The TL approach previews that the output of the CNN network is used throughout a specific algorithm for performing classification”. TL basically consists of using knowledge gained from solving one problem (i.e., knowledge from training data related to a certain problem) in making inferences about a different, but related, problem. Please rephrase more clearly.
o Line 72: please change “In the last years” to “In the last few years”, or “In the past # years”, or something along those lines
o Line 76: please change “bio-informatics” to “bioinformatics”
o Line 86: please change “builded” to “built”
o Lines 90-91: please change “The try also to provide a methodology for approaching to an accurate image classification problem” to “They try also to provide a methodology for approaching an accurate image classification problem”. Overall, the sentence is not clear, please consider rewriting it completely.
o Lines 136-137: please change “a correct and automatically granite identification” to “a correct and automated granite identification”
o Line 349: please change “that, is then as input of a classification layer” to “, which is then used as input to the classification layer “”
o Line 354: please change “have one layer more” to “have one more layer”
o Line 668: please change “How it is possible to observe” to “As it is possible to observe”
Round 2
Reviewer 1 Report
The revision is fine and the paper can be accepted.
Reviewer 3 Report
In my opinion the paper is now more coherent. Methodologies, conclusions and results are clearer. In particular, the abstract and Section 2.3 significantly clarify the main contributions of the paper. However, I still believe significant revisions of the English style are needed. Furthermore, there are some issues that need to be addressed before the paper is suitable for publication.
A suggestion for the authors: in future, in your written reply to a review, please when possible list the lines where you made the relevant modifications. That will significantly facilitate the revision.
Based on the above, in my opinion the paper is now closer to be suitable for publication, after implementing the relatively minor revisions listed below.
· I would merge the following contributions (Line 64-73), as both concern the application domain (stone classification):
o the paper context concerns the recognition of the main Calabrian quarries that, at the best of our knowledge, represents the first attempt in the stone literature
o the paper proposes a methodology to be used in the stone classification providing to the reader precise indications on this specific task
· Ref [33] seems to be the closest to your work since it employs CNNs, TL and various ML methods for microscopy image classification of minerals. If you agree, it probably deserves special attention. Could you explicitly mention the main differences between your work and theirs, for instance in Lines 171-177? My impression is that they might not have a fully implemented two-stage hybrid approach, or that they did not test their approach with several CNNs as you did. Please elaborate.
· I still believe Section 4, 4.1 and 4.2 are unnecessarily taking space. Image Recognition, Data Augmentation and Overfitting and CNNs do not need to be explained. Mentioning them and possibly citing references, for instance during the description of your method (Section 5), would suffice. Instead, why not to elaborate more on the choice of the 4 CNN models (Section 5.2). Why did you choose those four? Why did you choose those specific for the classification techniques (Section 5.3)? Sections 5.2 and 5.3 are relatively short, especially the latter, and could use some extra text to provide additional insight.
· Question on the augmentation (Section 7.3): why only cropping, and not also rotations? Any reason in particular?
· I still think Figg. 11-13 should be improved. You are discussing them and you are pointing the reader to specific entries of the matrix (Lines 504-508: “It possible to observe that the classes that present some recognition problems in all hybrid models are those related to the family of granite stones, that is GRB, GRD, GRS1, GRS2 and GRS3 families. This is due to a very similar texture of these types of stones that makes difficult also to an expert eye to capture the differences.”). It would greatly help to highlight off-diagonal non-zero entries. They are relatively few, so it should not be too challenging.
· Instances of style/grammar improvements, still needed.
o Line 90-91: please replace “In the following of this section” with “In the remainder of this section”
o Line 108-109: please replace “The proposal CNN” with “The proposed CNN”
o Line 173: please replace “Moreover, the feature extracted are” with “Moreover, the features extracted are”
